# Analysis of Volatile Components in Dried Fruits and Branch Exudates of *Schisandra chinensis* with Different Fruit Colors Using GC-IMS Technology

**DOI:** 10.3390/molecules28196865

**Published:** 2023-09-29

**Authors:** Yiping Yan, Wenpeng Lu, Taiping Tian, Nan Shu, Yiming Yang, Shutian Fan, Xianyan Han, Yunhua Ge, Peilei Xu

**Affiliations:** 1Institute of Special Animal and Plant Sciences of Chinese Academy of Agricultural Sciences, Changchun 130112, China; 82101225211@caas.cn (Y.Y.); luwenpeng@caas.cn (W.L.); 18685618868@163.com (T.T.); shunan@caas.cn (N.S.); yangyiming@caas.cn (Y.Y.); fanshutian@caas.cn (S.F.); hanxianyan@caas.cn (X.H.); geyunhua@caas.cn (Y.G.); 2Jilin Provincial Key Laboratory of Traditional Chinese Medicinal Materials Cultivation and Propagation, Changchun 130112, China

**Keywords:** *Schisandra chinensis*, HS-GC-IMS, volatile components, odor activity value, variable importance in projection

## Abstract

To investigate the volatile components of *Schisandra chinensis* (Turcz.) Bail (commonly known as northern *Schisandra*) of different colors and to explore their similarities and differences, to identify the main flavor substances in the volatile components of the branch exudates of northern *schisandra*, and finally to establish a fingerprint map of the volatile components of the dried fruits and branch exudates of northern *Schisandra* of different colors, we used GC-IMS technology to analyze the volatile components of the dried fruits and branch exudates of three different colors of northern Schisandra and established a fingerprint spectra. The results showed that a total of 60 different volatile chemical components were identified in the branch exudates and dried fruits of *Schisandra*. The components of germplasm resources with different fruit colors were significantly different. The ion mobility spectrum and OPLS-DA results showed that white and yellow fruits were more similar compared to red fruits. The volatile components in dried fruits were significantly higher than those in branch exudates. After VIP (variable importance in projection) screening, 41 key volatile substances in dried fruits and 30 key volatile substances in branch exudates were obtained. After screening by odor activity value (OAV), there were 24 volatile components greater than 1 in both dried fruits and branch exudates. The most important contributing volatile substance was 3-methyl-butanal, and the most important contributing volatile substance in white fruit was (*E*)-2-hexenal.

## 1. Introduction

*Schisandra chinensis* (Tuncz.) Baill. is a perennial woody vine of the *Schisandra* family mainly produced in eastern Russia, northeastern China, Korea, and other places. Its fruit is a traditional Chinese medicinal material, commonly known as “*Northern Schisandra*”. *Schisandra* contains a variety of medicinal active ingredients, mainly including lignans, polysaccharides, volatile oils, triterpenes, organic acids, etc. [1]. The 2020 edition of the Chinese Pharmacopoeia records 100 prescriptions related to *Schisandra*, which are usually used to treat various diseases, such as asthma and cardiovascular diseases [2,3]. The volatile oil of *Schisandra* can also improve cognitive dysfunction in mice [4]. The volatile chemical constituents of Schisandra, a significant part of which is Schisandra essential oil, have been extensively studied for their pharmacological activity. Experimental validation has shown that Schisandra essential oil has anti-inflammatory and neuroprotective effects [5]. The volatile components of Schisandra can significantly alleviate neuronal damage and apoptosis, protecting the hippocampal and cortical structures [6]. These volatile components also alleviate skeletal muscle cell damage in mice [7] and have shown potential resistance to atherosclerosis [8]. Schisandra not only has a pleasant aroma but also has high potential pharmacological activity, making it of great research value in both culinary and medicinal applications. The dried fruits of Schisandra chinensis contain about 20% volatile oil, which mainly consists of terpene compounds [9].

Gas chromatography–ion mobility spectrometry (GC-IMS) is a sensitive and rapid gas phase separation detection technology that has emerged in recent years. It combines ion mobility spectrometry with gas chromatography to generate a two-dimensional spectrum of volatile compounds. This is based on the retention characteristics of the gas chromatography column and the ion mobility rate of the ion mobility spectrometry detector, providing a more convenient, faster, and accurate method of analysis [10]. Furthermore, the sample does not require complex concentration and enrichment, which helps maintain the stability of flavor substances. As such, GC-IMS can be widely used for distinguishing volatile components and isomers, analyzing trace components, and facilitating rapid on-site detection [11]. It is commonly used for odor detection in various fields, such as food and environmental pollution, and is very suitable for the rapid detection of volatile components [12]. GC-IMS technology can be applied to the identification of volatile organic compounds in traditional Chinese medicine materials, and the measurement results can be used as a reference for quality evaluation and variety selection. The University of Chinese Academy of Sciences applied GC-IMS technology to measure the effect of different drying methods on the volatile oil components of wolfberry [13]; Tianjin University of Traditional Chinese Medicine used GC-IMS technology to study the fingerprint of Asarum volatile oil and its chemical pattern recognition [14]; Zhejiang Institute of Food and Drug Inspection established a fingerprint of volatile substances in Aurantium using GC-IMS technology and established a new method for identifying Aurantium [15]. Heilongjiang University of Traditional Chinese Medicine applied GC-IMS technology to measure the volatile oil components of *Schisandra*, and the results showed that the composition of *Schisandra* volatile components was complex, including terpenes, aromatics, aliphatics, and other compounds [6].

According to the “Flora of China”, *Schisandra* is a small red berry that is nearly spherical or obovate in shape, with a diameter of 6–8 mm. Most *Schisandra* berries are red or dark red, and people often use the depth of the red color as a sensory indicator to evaluate the quality of *Schisandra*. As a result, the color of *Schisandra* berries has been subjected to directional selection pressure during natural evolution. White and yellow germplasms, as freely segregating offspring, are very precious germplasm resources. The *Schisandra* National Forest Germplasm Bank of the Institute of Special Products of the Chinese Academy of Agricultural Sciences has collected white and yellow *Schisandra* germplasm resources and used SRAP technology to analyze the genetic rules of fruit color. The results show that the genotype of red *Schisandra* is dominant homozygous, white is recessive homozygous, and yellow fruit is heterozygous [16]. However, there have been no reports on the differences in volatile components between different fruit colors. In this study, we measured the volatile components of dried fruits and branch exudates of three different fruit colors of *Schisandra* and established a fingerprint map of volatile components of different fruit colors of *Schisandra*. Using principal component analysis, cluster analysis, OAV value analysis, OPLS-DA analysis and other methods, we screened the key aroma components of *Schisandra* and qualitatively analyzed the key volatile components of different fruit colors of *Schisandra* fruits. This lays a foundation for germplasm identification, quality evaluation, variety selection, and development of volatile compounds in dried fruits and branch exudates.

## 2. Results and Discussion

### 2.1. Analysis of Ion Migration Spectra of Schisandra chinensis Branch Sap and Dried Fruits of Different Colors

Using GC-IMS technology to analyze the volatile components of branch sap and dried fruits of three different colors, the two-dimensional ion migration spectra of volatile organic compounds obtained are shown in Figure 1 and Figure 2. The horizontal axis represents the ion migration time, and the vertical axis represents the gas chromatographic retention time. The red vertical line at 1.0 on the horizontal axis is the RIP (reaction ion peak), and the dots shown in the picture are the volatile organic compounds detected in the sample. From white to red, the deeper the color, the higher the relative concentration of volatile organic compounds.

#### 2.1.1. Ion Migration Spectra Analysis of White, Yellow, and Red Fruit Branch Sap

Figure 1 shows that the peak positions of W1 and W2 in the three ion migration spectra are close, and the composition of W3 is significantly different from that of W1 and W2. Taking W1 as a reference and subtracting the signal peaks of W1 in the remaining spectra, a differential spectrum can be obtained. In the differential spectrum, blue dots indicate that the corresponding volatile organic compounds in this sample are reduced compared to those at the same position in W1. Red dots indicate that this substance contains more than W1, and the deeper the color, the greater the difference. In the differential spectrum, it can be found that compared with W1, except for some positions where the content of W2 is reduced, there are fewer volatile organic compound difference points. The content difference of volatile organic compounds in W3 is more significant, and there are more difference substances compared with W1 and W2.

#### 2.1.2. Ion Migration Spectra Analysis of White, Yellow, and Red *Schisandra chinensis* Dried Fruits

The ion migration spectra of dried fruits of different colors are shown in Figure 2. It can be seen that there are more points separated in the dried fruit spectra than in the branch sap spectra. By judging the peak positions, it can be preliminarily seen that the components of W4 and W5 are more similar and significantly different from those of W6. The differential spectrum shows that there are more blue dots in W6, indicating that the relative content of some volatile chemical substances in W4 and W5 is higher than that in W6.

### 2.2. Qualitative Analysis of Volatile Chemical Substances in White, Yellow, and Red Fruit Branch Sap and Dried Fruits

The VoCal software 0.4.03built into the flavor analyzer is used to analyze the ion migration spectra. The NIST and IMS databases built into the software are used to organize and compare the original data in the GC-IMS spectra to obtain the fingerprint spectra of the sample’s volatile chemical substances. The experiment uses 4-methyl-2-pentanol as an internal standard. By comparing the actual retention index and ion drift time of different substances with the database, it is possible to compare volatile chemical substances. The peak volume of volatile chemical substances is calculated, and the concentration of chemical substances can be obtained by calculating using the internal standard method, achieving quantification of chemical substances in different samples. Volatile compounds that are not identified are represented by numerical codes.

As shown in Figure 3 and Figure 4, the difference in the content of each substance can be intuitively seen from the depth of the colors on the fingerprint spectra. The establishment of fingerprint spectra visualizes the spectral data and directly helps in the directional selection of germplasm resources rich in special chemical substances (odor thresholds: compilations of odor threshold values in air, water, and other media (edition 2011)). Fingerprint spectra can also be used as a reference for the evaluation and identification of *Schisandra chinensis* germplasm resources in the future. Consistent with the results of the migration nursery, the number and types of volatile substances in *Schisandra chinensis* dried fruits are significantly higher than those in *Schisandra chinensis* branch sap, indicating that the aroma of *Schisandra chinensis* dried fruits is richer than that of branch sap. W1 and W2 have high similarity, W4 and W5 have high similarity, and the fingerprint spectra of white and yellow fruits are more similar.

### 2.3. Classification of Volatile Chemical Substances in Branch Sap and Dried Fruits

The volatile compounds in dried fruits and branch exudates can be classified into alkenes, ketones, aldehydes, alcohols, esters, and others (as shown in Figure 5). Overall, the amount of volatile substances detected in dried fruits is higher than that in branch exudates. Among them, alkenes, alcohols, ketones, and aldehydes are present in high amounts in dried fruits. The volatile components in branch exudates are mainly alcohols, ketones, and aldehydes. Alkenes are the high-content volatile substances in dried fruits. The total amount of volatile substances in white and yellow fruits is relatively high, both in dried fruits and branch exudates. The content of alkenes is the highest in W4 and W5. In the corresponding W1 and W2, the content of ketones is the highest, followed by alcohols. The content of alcohols is relatively high in W3. The qualitative analysis results are shown in Table 1. A total of 83 volatile compounds were identified.

### 2.4. Principal Component Analysis and OPLS-DA Analysis of Samples

The peak volume data of monomers and dimers of the same chemical substance detected were merged and the experimental data obtained from six injections were analyzed by principal component analysis (PCA-X), orthogonal partial least squares discriminant analysis (OPLS-DA) and 200 times permutation test using SIMCA 14.0 software. The branch exudates and *Schisandra* dried fruits were divided into two groups, and the VIP values of each component were calculated. SPSS 27 software was used to perform one-way ANOVA analysis and homogeneity of variance test on the chemical components of branch exudates and *Schisandra* dried fruits, respectively, to obtain *p* values. Volatile components with special contributions were screened using *p* value < 0.05 and VIP value > 1 as the criteria for further data analysis.

A total of 60 different volatile chemical components were identified in *Schisandra* branch exudates and dried fruits using GC-IMS. With 60 common components as dependent variables and dried fruits, branch exudates and different fruit colors as independent variables divided into six groups, PCA-X and OPLS-DA analysis of samples of *Schisandra* dried fruits and branch exudates of different colors can achieve effective differentiation.

The volatile chemical components obtained were processed using PCA, resulting in the data shown in Figure 6. The R2X (1) = 0.648 and the R2X (2) = 0.13, indicating that the model samples have good reliability. It can be observed that there is a good distinction between the dried fruits and branch exudates, and the differences between different groups are also noticeable. The data obtained from PCA was further processed and analyzed. In the OPLS-DA analysis (Figure 7 and Figure 8), the independent variable fitting index R2X = 0.972, the dependent variable fitting index R2Y = 0.992, and the model prediction index Q2 = 0.965 are all greater than 0.5, and the fitting result is acceptable [17]. Then, 200 permutation analyses were performed on all experimental data to verify whether the model is effective (Figure 9). As shown in the figure above, the intersection of the Q2 regression line with the vertical axis is less than 0, indicating that the model is effective, that there is no overfitting, and that the analysis is effective.

The results show that there are significant differences in the main component composition of *Schisandra* of different colors. The biplot shows that the dried fruit samples are more closely related to more volatile components, indicating that the flavor of *Schisandra* dried fruits is richer and the main flavor components of branch exudates are significantly different from those of dried fruits.

*Schisandra* branch exudates and dried fruits were divided into two groups and OPLS-DA analysis was performed separately. The VIP values of the odor components of *Schisandra* branch exudates and dried fruits were calculated separately to screen key volatile components. At the same time, SPSS software was used to perform one-way ANOVA analysis to calculate the significance of single components. Lists with *p* < 0.05 and VIP > 1 were screened separately for further discussion(Table 2).

The screening results show that there are 30 key volatile substances in branch exudates and 41 key volatile substances in dried fruits in terms of content. The VIP values show that the chemical substances with the greatest contribution in the branch exudate group are: 2-ethylhexanol (1.234), ethanol (1.219), acetone (1.193), 2-hexanone (1.189), and 1-propanol (1.186); in the dried fruit group, the key substances with the greatest contribution are: (*E*)-2-heptenal (1.159), 5-methylfurfural (1.158), cyclohexanone (1.154), (*E*)-2-hexenal (1.153), and 2-acetylfuran (1.153). The results indicate that the aroma composition of *Schisandra* dried fruits and branch exudates is complex, with many chemical substances contributing significantly, and there is a significant difference between the volatile substances of branch exudates and dried fruits.

In order to further visually display the contribution of different aroma component contents to different samples, ORIGIN was used to perform differential analysis on the aroma of *Schisandra* branch exudates and dried fruits. According to the criteria of *p* < 0.05 and VIP > 1, flavor substances were screened and classified [18]. The peak area was normalized, and cluster heat maps were made separately. The red part in the figure can directly show the classification according to the different component contents and can also see the highest and most prominent volatile substances (Figure 10 and Figure 11).

### 2.5. Analysis of OAV Values of Schisandra chinensis Branch Exudates and Dried Fruit and Screening of Key Flavor Substances

VIP values can only indicate the contribution of volatile components, and do not fully represent the composition of odors. The true olfactory composition of *Schisandra chinensis* branch exudates and dried fruit cannot be determined solely by VIP values. The introduction of odor activity values (OAV) is necessary to determine the aroma characteristics of *Schisandra chinensis* branch exudates and dried fruit. Calculating OAV values can more intuitively show the contribution of a single component to the overall aroma. It is generally believed that: OAV > 1 can be considered to have a certain impact on the aroma of the sample, and OAV > 10 proves that the aroma component has a great impact on the aroma of the sample. Calculate the OAV values and standard deviations of each aroma component in the branch exudates and *Schisandra chinensis* dried fruit, and screen according to the standard of OAV > 1, and then perform correlation analysis.

The internal standard method is used to quantify the chemical substances detected in six groups of samples. The internal standard is 4-methyl-2-pentanol with a concentration of 198 ppb and a signal peak volume of 504.95. Therefore, the intensity of each signal peak is about 0.392 ppb. The detected chemical substances are analyzed and calculated to finally obtain the concentration and standard deviation of three repeated injections of each chemical substance. *Schisandra chinensis* dried fruit selects the air threshold, and *Schisandra chinensis* branch exudates selects the water threshold. A total of 57 components can query the threshold value and calculate according to the method described above.

After screening, W1 has 15 aroma components with OAV values greater than 1 and 8 aroma components greater than 10; W2 has 15 aroma components with OAV values greater than 1 and 8 aroma components greater than 10; W3 has 14 aroma components with OAV values greater than 1 and 8 aroma components greater than 10; W4 has 24 aroma components with OAV values greater than 1 and 12 aroma components greater than 10; W5 has a total of 22 aroma components with OAV values greater than 1 and 14 aroma components greater than 10; W6 has a total of 22 aroma components with OAV values greater than 1 and 10 aroma components greater than 10. Except for the white dried fruit, the main contributing aroma is (*E*)-2-hexenal, and the main contributing aroma of the remaining samples is all 3-methyl-butanal (Table 3).

### 2.6. Discussion

Through the screening of OAV values, the key flavor substance in white fruit dried fruit is (*E*)-2-hexenal. This substance is also the main aroma-contributing substance of Chinese jujubes, with a strong plant aroma (fresh green, multi-leaf, rich fruit flavor) flavor characteristics [4] (http://www.thegoodscentscompany.com/search2.html, accessed on 28 March 2023). Both *Schisandra chinensis* dried fruit and branch exudates show that 3-Methyl-butanal is the key flavor substance of *Schisandra chinensis*. This substance is the main aroma component of green tea that has been killed and has elegant peach, chocolate, and fat aromas [19]. It can be used as a characteristic volatile label substance for Schisandra chinensis identification. 1,8-cineole in dried fruit is also a major contributing aroma substance with a cool mint and camphor smell. It is the main aroma component of Australian grape wine and is also used as a food additive and spice [20]. At the same time, chemical substances such as β-ocimene and myrcene with high OAV values have also been found in traditional Chinese medicines such as qianhu and have obvious aromatic odors (Surendran et al. 2021; Jovanović et al. 2015 [21]). There are a total of 24 volatile chemical substances with OAV > 1 in *Schisandra chinensis* dried fruit and branch exudates. The aroma substances are rich and have high potential for tea, seasoning, and spice development.

## 3. Materials and Methods

### 3.1. Experimental Materials

Six-year-old *Schisandra* trees were used as materials. White fruit germplasm (3N2S2), yellow fruit germplasm (variety “Jinwuwei No. 1”) and red fruit germplasm (variety “Yanzhihong”) were all collected from the *Schisandra* National Forest Germplasm Resource Bank of the Institute of Special Products of the Chinese Academy of Agricultural Sciences (Figure 12). Three clonal plants were selected for each resource. The three different-colored *Schisandra* fruits were harvested when they were fully ripe in September 2019. After removing the stems and branches in the laboratory, they were dried in the shade at 22 °C. They were completely dried after one month and then vacuum-sealed and stored in a refrigerator at 4 °C, respectively. In March of the following spring, branch exudates from the above germplasms were collected, placed in 50 mL centrifuge tubes, and labeled as white juice, yellow juice, and red juice. Three parallel samples were set up for each sample. W1, W2, and W3 are branch exudates of white-fruit, yellow-fruit, and red-fruit *Schisandra*, respectively. W4, W5, and W6 are dried fruits of white-fruit, yellow-fruit, and red-fruit *Schisandra*, respectively.

### 3.2. Experimental Methods

#### 3.2.1. Sample Pre-Treatment

Take 1 mL of fresh branch exudate and place it in a 20 mL headspace vial. Incubate at 50 °C for 15 min before sampling. Take 1 g of the sample and crush it, then place it in a 20 mL headspace vial. Incubate at 50 °C for 15 min before sampling.

#### 3.2.2. GC-IMS Conditions

The FlavourSpec^®^ flavor analyzer (GAS Company, Jinan, China) was used in the experiment. The operating conditions of the detector were as follows: input voltage: 220 V, frequency 50 Hz, ambient temperature 22 °C, humidity 42%, and the gas used was high-purity nitrogen (99.999%). The detector’s ionization source was a tritium source (H^3^), with the type of radiation being beta radiation. The radiation energy of the ionization source was 6.5 KeV. The length of the drift tube was 98 mm, and the voltage of the drift tube was 5000 V.

Gas phase–ion mobility spectrometry unit: analysis time 25 min; column type: MXT-5, length 15 m, inner diameter 0.53 mm, film thickness 1 μm; column temperature 60 °C; carrier gas/drift gas: N_2_; IMS temperature: 45 °C.

Automatic headspace sampler unit: sample volume 400 μL; incubation time: 15 min; incubation temperature: 50 °C; sample needle temperature: 55 °C; incubation speed: 500 rpm.

The initial carrier gas flow rate is set to 2 mL/min for 0–2 min, and the carrier gas flow rate is increased to 100 mL/min for 2–20 min. The carrier gas flow rate is maintained at 100 mL/min for 20–25 min until the end.

#### 3.2.3. Data Processing Conditions

Preliminary analysis: Use the VOCal analysis software that comes with the FlavourSpec^®^ flavor analyzer. The VOCal software can be used to analyze the spectra output from the instrument and perform qualitative and quantitative data analysis. The built-in IMS and NIST databases can be used to compare and qualitatively identify volatile substances. After establishing a standard curve for the identified substances, quantitative analysis can be performed. The database is equipped with three plug-ins: Reporter, Gallery Plot, and Dynamic PCA. The Reporter plug-in can directly compare spectral differences in two-dimensional, three-dimensional and other aspects; the Gallery Plot plug-in can compare the output fingerprint spectra and quantitatively and intuitively compare the differences in volatile substances between different samples; the Dynamic PCA plug-in can perform dynamic principal component analysis and similarity analysis to quickly identify unknown samples [22].

Quantitative calculation of chemical components:(1)$$Ci=\frac{Cis\times Ai}{Ais}$$
*Ci* is the calculated mass concentration of the chemical component, in μg/L; *Cis* is the mass concentration of the internal standard substance. The internal standard substance used in the experiment is 4-methyl-2-pentanol, with a concentration of 198 μg/L; *Ai* is the signal peak volume of the chemical component; *Ais* is the signal peak volume of the internal standard substance.

#### 3.2.4. OAV Value Analysis

The OAV analysis method will be used to analyze and screen the main aroma components. The calculation of the OAV value is related to the threshold value of the main aroma component itself. The selection of the threshold value mainly refers to the book “ODOUR THRESHOLDS”. Due to different years and different statistical methods, there may be differences in the same substance. The principle for selecting the threshold value in this experiment is: the air threshold value is selected for *Schisandra* dried fruit, and the water threshold value is selected for *Schisandra* branch exudate. All threshold values are selected from the latest data (odor thresholds: compilations of odor threshold values in air, water and other media).

Calculation of OAV value:(2)$$OAV=\frac{C}{OT}$$
*C*—the amount of substance, in units of mg/m^3^ (in air), mg/kg (in water); *OT*—the threshold value of the substance itself.

#### 3.2.5. OPLS-DA Analysis

OPLS-DA analysis, also known as orthogonal partial least squares discriminant analysis, is used to observe the clustering of samples based on principal component analysis (PCA) data analysis. The OPLS-DA analysis model is then used to screen VIP values, identify differential variables, and locate key differential volatile substances. This part was used by SIMCA and SPSS.

## 4. Conclusions

This study applied GC-IMS technology for the first time to analyze volatile chemical substances in *Schisandra chinensis* branch exudates and dried fruit of different colors. The volatile substance fingerprint of white-, yellow-, and red-fruit *Schisandra chinensis* branch exudates and dried fruit was established. This fingerprint can effectively distinguish *Schisandra chinensis* dried fruit and branch exudates of different colors and can be directly used for quality characteristic evaluation and germplasm identification.

Different algorithms, such as PCA-X, one-way ANOVA, OPLS-DA, and OAV were used to compare and analyze the volatile components of *Schisandra chinensis*. The results showed significant differences in content among the aroma components of *Schisandra chinensis* branch exudates and dried fruit. The content and number of high-content types of aroma components in *Schisandra chinensis* dried fruit are higher than those in branch exudate samples. Comparing between different fruit colors, it was found that the aroma composition of white fruit and yellow fruit *Schisandra chinensis* is closer, with red fruit *Schisandra chinensis* significantly different from yellow fruit and white fruit.

The experiment screened out key volatile chemical substances of dried fruits and branch exudates through VIP value and *p* value joint screening. The key volatile substances of overripe fruits are more abundant than those of branch exudates, with complex composition and significant differences. These findings highlight the importance of considering both plant part and maturity stage in analyzing volatile components, providing valuable insights for future research in plant chemistry.

## Figures and Tables

**Figure 1 molecules-28-06865-f001:**
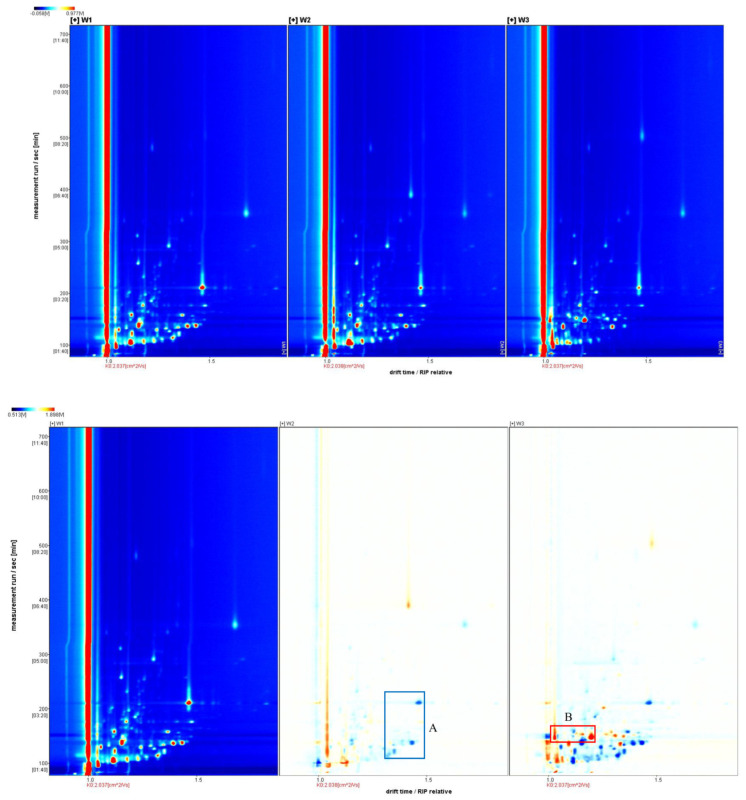
Two-dimensional ion migration spectra (**top**) and differential spectra (**bottom**) of white, yellow, and red *Schisandra chinensis* branch sap. A is the substance that is less in the W2 sample compared to the other samples, B is the substance that is increased in the W3 sample compared to the other samples.

**Figure 2 molecules-28-06865-f002:**
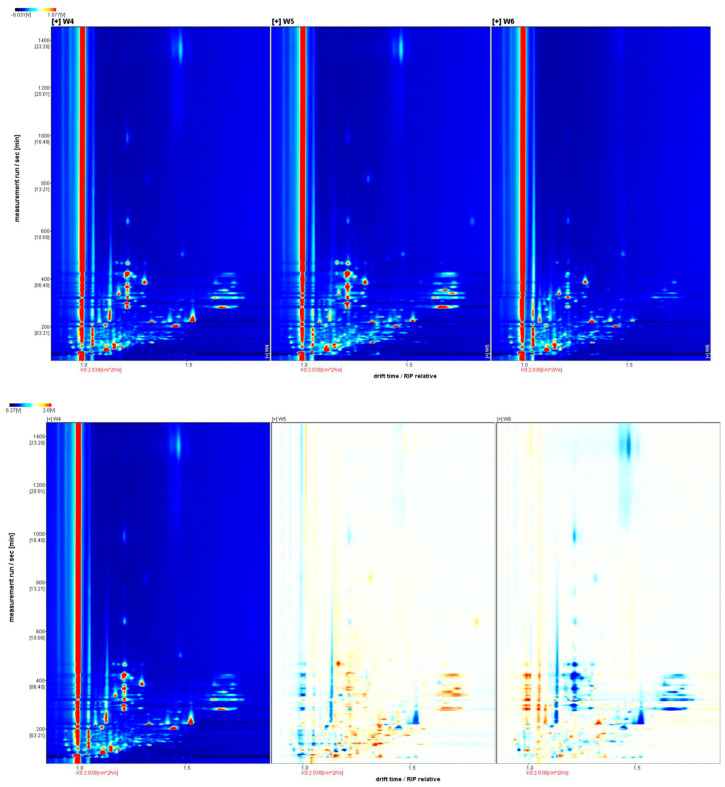
Twodimensional ion migration spectra (**top**) and differential spectra (**bottom**) of white, yellow, and red *Schisandra chinensis* dried fruits.

**Figure 3 molecules-28-06865-f003:**
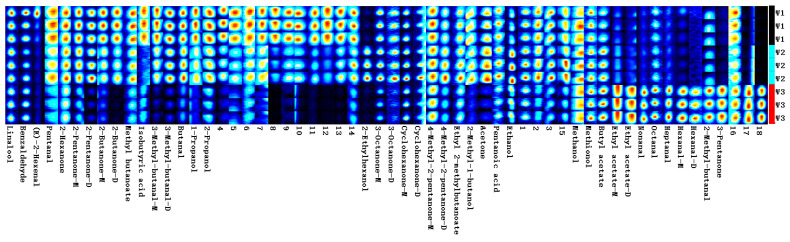
Fingerprint spectra of white, yellow, and red *Schisandra chinensis* branch sap.

**Figure 4 molecules-28-06865-f004:**
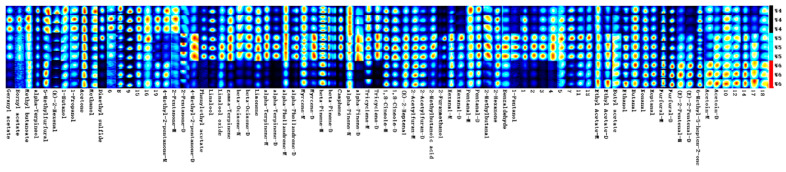
Fingerprint spectra of white, yellow, and red *Schisandra chinensis* dried fruits.

**Figure 5 molecules-28-06865-f005:**
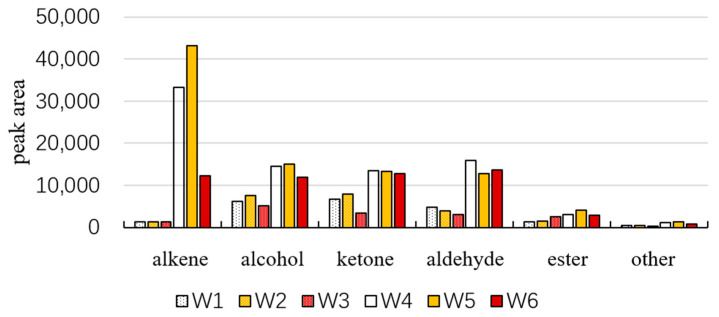
Peak area of different types of compounds in *Schisandra*.

**Figure 6 molecules-28-06865-f006:**
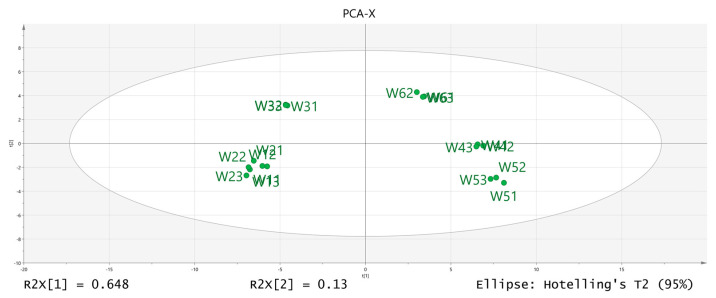
Results of PCA-X (principal component analysis) analysis.

**Figure 7 molecules-28-06865-f007:**
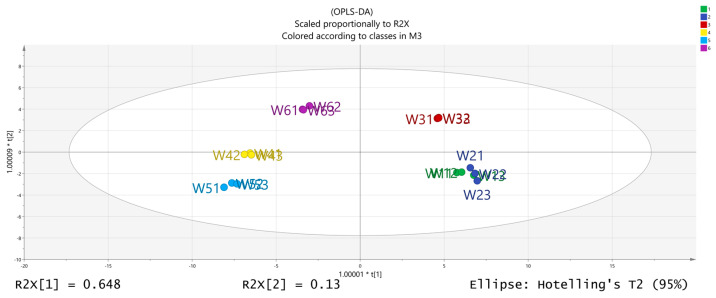
Results of OPLS-DA (orthogonal partial least squares discriminant analysis) analysis.

**Figure 8 molecules-28-06865-f008:**
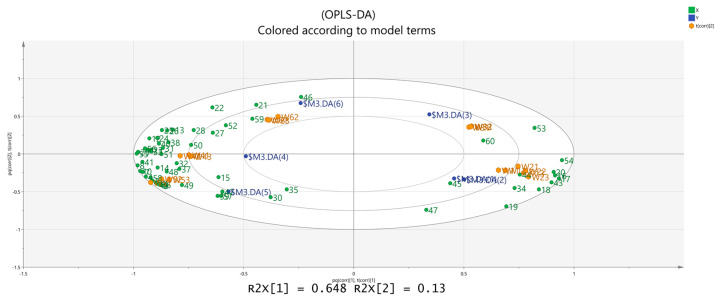
Biplot of OPLS-DA analysis.

**Figure 9 molecules-28-06865-f009:**
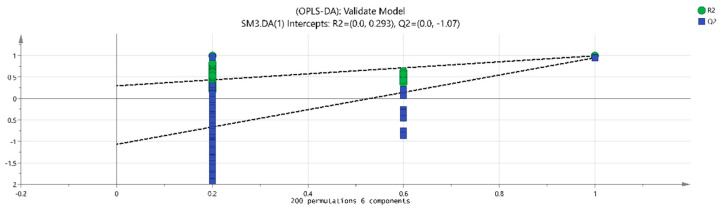
Permutation retention.

**Figure 10 molecules-28-06865-f010:**
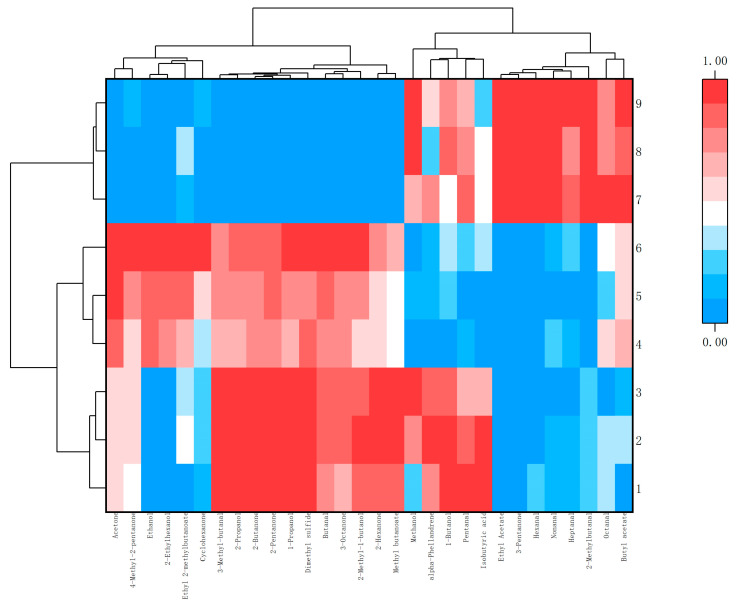
Cluster heat map of differential aroma components in branch exudates.

**Figure 11 molecules-28-06865-f011:**
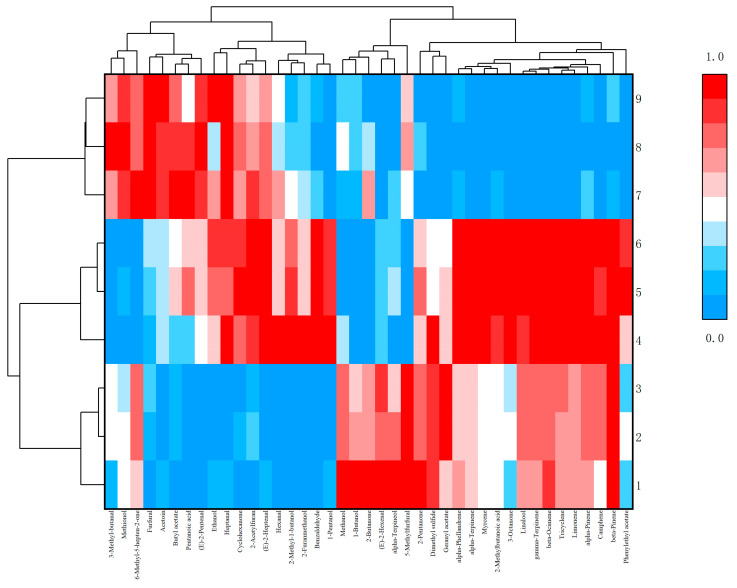
Cluster heat map of differential aroma components in dried fruits.

**Figure 12 molecules-28-06865-f012:**
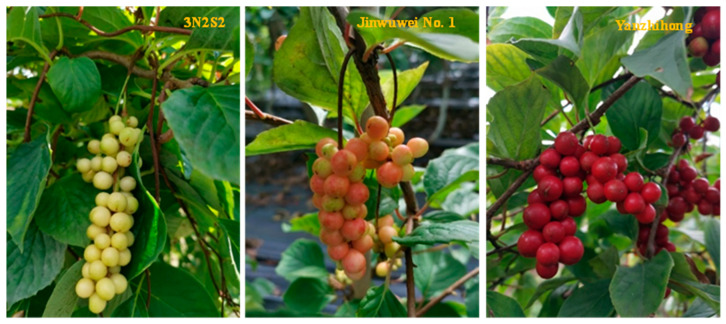
White, yellow, and red *Schisandra* fruits.

**Table 1 molecules-28-06865-t001:** Volatile chemical substances detected in samples.

No.	Compound	CAS#	Formula	Molecular Weight	Retention Index	tR (s)	Migration Time (ms)	Note
1	α-Terpineol	C98555	C_10_H_18_O	154.3	1205.4	641.106	1.22046	
2	γ-Terpinene	C99854	C_10_H_16_	136.2	1054.0	423.305	1.2173	
3	β-Ocimene	C13877913	C_10_H_16_	136.2	1045.0	410.429	1.21073	Monomer
4	β-Ocimene	C13877913	C_10_H_16_	136.2	1044.5	409.742	1.24999	Dimer
5	Limonene	C138863	C_10_H_16_	136.2	1024.9	381.459	1.2184	
6	α-Terpinene	C99865	C_10_H_16_	136.2	1014.8	366.973	1.2173	Monomer
7	α-Terpinene	C99865	C_10_H_16_	136.2	1014.7	366.844	1.72557	Dimer
8	α-Phellandrene	C99832	C_10_H_16_	136.2	1005.1	352.997	1.22057	Monomer
9	α-Phellandrene	C99832	C_10_H_16_	136.2	1005.3	353.257	1.68616	Dimer
10	Myrcene	C123353	C_10_H_16_	136.2	995.7	340.021	1.21844	Monomer
11	Myrcene	C123353	C_10_H_16_	136.2	996.3	340.54	1.7223	Dimer
12	β-Pinene	C127913	C_10_H_16_	136.2	973.5	321.596	1.21738	Monomer
13	β-Pinene	C127913	C_10_H_16_	136.2	973.5	321.596	1.63939	Dimer
14	Camphene	C79925	C_10_H_16_	136.2	945.3	298.239	1.21419	
15	α-Pinene	C80568	C_10_H_16_	136.2	932.5	287.599	1.21738	Monomer
16	α-Pinene	C80568	C_10_H_16_	136.2	932.2	287.34	1.6649	Dimer
17	Tricyclene	C508327	C_10_H_16_	136.2	922.5	279.295	1.21632	Monomer
18	Tricyclene	C508327	C_10_H_16_	136.2	924.1	280.592	1.67234	Dimer
19	6-Methyl-5-hepten-2-one	C110930	C_8_H_14_O	126.2	988.9	334.35	1.1776	
20	Acetoin	C513860	C_4_H_8_O_2_	88.1	724.7	177.155	1.07174	Monomer
21	Acetoin	C513860	C_4_H_8_O_2_	88.1	724.7	177.155	1.33376	Dimer
22	Acetone	C67641	C_3_H_6_O	58.1	447.9	108.943	1.11495	
23	2-Butanone	C78933	C_4_H_8_O	72.1	541.0	129.96	1.05779	Monomer
24	2-Butanone	C78933	C_4_H_8_O	72.1	541.0	129.96	1.24512	Dimer
25	2-Hexanone	C591786	C_6_H_12_O	100.2	776.5	197.058	1.18758	
26	4-Methyl-2-pentanone	C108101	C_6_H_12_O	100.2	722.8	176.431	1.17655	Monomer
27	4-Methyl-2-pentanone	C108101	C_6_H_12_O	100.2	722.3	176.25	1.47994	Dimer
28	2-Pentanone	C107879	C_5_H_10_O	86.1	665.2	158.011	1.12146	Monomer
29	2-Pentanone	C107879	C_5_H_10_O	86.1	669.1	158.902	1.36775	Dimer
30	3-Octanone	C106683	C_8_H_16_O	128.2	936.0	290.5	1.30341	Monomer
31	3-Octanone	C106683	C_8_H_16_O	128.2	937.3	291.569	1.71458	Dimer
32	Cyclohexanone	C108941	C_6_H_10_O	98.1	897.9	258.855	1.15475	Monomer
33	Cyclohexanone	C108941	C_6_H_10_O	98.1	898.4	259.264	1.45434	Dimer
34	3-Pentanone	C96220	C_5_H_10_O	86.1	629.9	150.028	1.35114	
35	Nonanal	C124196	C_9_H_18_O	142.2	1109.8	503.569	1.48117	
36	(*E*)-2-Heptenal	C18829555	C_7_H_12_O	112.2	112.2	307.472	1.25622	
37	5-Methylfurfural	C620020	C_6_H_6_O_2_	110.1	965.2	314.693	1.12757	
38	Heptanal	C111717	C_7_H_14_O	114.2	901.8	262.107	1.33019	
39	Furfural	C98011	C_5_H_4_O_2_	96.1	824.9	220.414	1.08181	Monomer
40	Furfural	C98011	C_5_H_4_O_2_	96.1	824.9	220.414	1.33019	Dimer
41	Hexanal	C66251	C_6_H_12_O	100.2	787.7	201.762	1.25745	Monomer
42	Hexanal	C66251	C_6_H_12_O	100.2	785.9	200.857	1.56268	Dimer
43	(*E*)-2-Pentenal	C1576870	C_5_H_8_O	84.1	743.1	184.211	1.10576	Monomer
44	(*E*)-2-Pentenal	C1576870	C_5_H_8_O	84.1	740.4	183.208	1.35645	Dimer
45	Pentanal	C110623	C_5_H_10_O	86.1	679.5	161.233	1.18206	Monomer
46	Pentanal	C110623	C_5_H_10_O	86.1	682.7	161.957	1.42202	Dimer
47	Butanal	C123728	C_4_H_8_O	72.1	554.5	133.007	1.29147	
48	Octanal	C124130	C_8_H_16_O	128.2	1005.0	352.965	1.41192	
49	(*E*)-2-Hexenal	C6728263	C_6_H_10_O	98.1	849.4	232.684	1.17887	Monomer
50	(*E*)-2-Hexenal	C6728263	C_6_H_10_O	98.1	834.7	225.319	1.52791	Dimer
51	2-Methylbutanal	C96173	C_5_H_10_O	86.1	641.9	152.737	1.15934	Monomer
52	2-Methylbutanal	C96173	C_5_H_10_O	86.1	635.4	151.282	1.3972	Dimer
53	3-Methyl-butanal	C590863	C_5_H_10_O	86.1	580.5	138.866	1.15386	Monomer
54	3-Methyl-butanal	C590863	C_5_H_10_O	86.1	576.1	137.877	1.39403	Dimer
55	Benzaldehyde	C100527	C_7_H_6_O	106.1	959.5	309.982	1.14884	
56	Linalool	C78706	C_10_H_18_O	154.3	1083.8	466.225	1.2173	
57	2-Furanmethanol	C98000	C_5_H_6_O_2_	98.1	867.1	241.567	1.37689	
58	1-Butanol	C71363	C_4_H_10_O	74.1	643.4	153.091	1.17931	
59	1-Propanol	C71238	C_3_H_8_O	60.1	536.9	129.027	1.2455	
60	Methanol	C67561	CH_4_O	32.0	370.2	91.393	0.9844	
61	Linalool oxide	C60047178	C_10_H_18_O_2_	170.3	1069.3	445.374	1.2601	
62	1-Pentanol	C71410	C_5_H_12_O	88.1	756.0	189.171	1.51258	
63	2-Propanol	C67630	C_3_H_8_O	60.1	455.6	110.664	1.22879	
64	Ethanol	C64175	C_2_H_6_O	46.1	401.5	98.449	1.04508	
65	2-Ethylhexanol	C104767	C_8_H_18_O	130.2	1031.3	390.652	1.4199	
66	Methionol	C505102	C_4_H_10_OS	106.2	996.0	340.217	1.08499	
67	2-Methyl-1-butanol	C137326	C_5_H_12_O	88.1	759.6	190.57	1.23167	
68	Geranyl acetate	C105873	C_12_H_20_O_2_	196.3	1448.3	990.472	1.21909	
69	Phenylethyl acetate	C103457	C_10_H_12_O_2_	164.2	1327.5	816.675	1.3188	
70	Bornyl acetate	C76493	C_12_H_20_O_2_	196.3	1289.3	761.699	1.21909	
71	Methyl butanoate	C623427	C_5_H_10_O_2_	102.1	738.3	182.402	1.14989	
72	Ethyl Acetate	C141786	C_4_H_8_O_2_	88.1	564.9	135.36	1.09381	Monomer
73	Ethyl Acetate	C141786	C_4_H_8_O_2_	88.1	570.5	136.626	1.33468	Dimer
74	Butyl acetate	C123864	C_6_H_12_O_2_	116.2	800.7	208.276	1.23539	
75	Ethyl 2-methylbutanoate	C7452791	C_7_H_14_O_2_	130.2	837.1	226.551	1.23473	
76	1,8-Cineole	C470826	C_10_H_18_O	154.3	1026.4	383.613	1.30141	Monomer
77	1,8-Cineole	C470826	C_10_H_18_O	154.3	1026.0	383.045	1.72693	Dimer
78	2-Acetylfuran	C1192627	C_6_H_6_O_2_	110.1	912.0	270.566	1.11566	Monomer
79	2-Acetylfuran	C1192627	C_6_H_6_O_2_	110.1	912.5	270.998	1.43948	Dimer
80	Dimethyl sulfide	C75183	C_2_H_6_S	62.1	475.2	115.095	0.95774	
81	2-Methylbutanoic acid	C116530	C_5_H_10_O_2_	102.1	901.4	261.801	1.20302	
82	Isobutyric acid	C79312	C_4_H_8_O_2_	88.1	757.0	189.581	1.16251	
83	Pentanoic acid	C109524	C_5_H_10_O_2_	102.1	888.2	252.17	1.23071	

**Table 2 molecules-28-06865-t002:** Different volatile components of *Schisandra* branch exudates and dried fruits of different colors.

No.	Compound (Juice)	*p*	VIP	No.	Compound (Fruit)	*p*	VIP
1	α-Phellandrene	0.0039	1.115	1	α-Terpineol	0.0112	1.040
2	Acetone	0.0000	1.193	2	γ-Terpinene	0.0000	1.058
3	2-Butanone	0.0000	1.181	3	β-Ocimene	0.0000	1.068
4	2-Hexanone	0.0000	1.189	4	Limonene	0.0000	1.061
5	4-Methyl-2-pentanone	0.0029	1.150	5	α-Terpinene	0.0000	1.061
6	2-Pentanone	0.0000	1.180	6	α-Phellandrene	0.0000	1.061
7	3-Octanone	0.0001	1.178	7	Myrcene	0.0000	1.069
8	Cyclohexanone	0.0325	1.072	8	β-Pinene	0.0000	1.070
9	3-Pentanone	0.0000	1.173	9	Camphene	0.0001	1.032
10	Nonanal	0.0000	1.182	10	α-Pinene	0.0000	1.051
11	Heptanal	0.0000	1.153	11	Tricyclene	0.0000	1.061
12	Hexanal	0.0000	1.167	12	6-Methyl-5-hepten-2-one	0.0007	1.034
13	Pentanal	0.0014	1.113	13	Acetoin	0.0000	1.098
14	Butanal	0.0000	1.184	14	2-Butanone	0.0105	1.040
15	Octanal	0.0063	1.124	15	2-Pentanone	0.0010	1.067
16	2-Methylbutanal	0.0000	1.174	16	3-Octanone	0.0000	1.067
17	3-Methyl-butanal	0.0000	1.185	17	Cyclohexanone	0.0002	1.154
18	1-Butanol	0.0046	1.066	18	(*E*)-2-Heptenal	0.0000	1.159
19	1-Propanol	0.0000	1.186	19	5-Methylfurfural	0.0000	1.158
20	Methanol	0.0213	1.051	20	Heptanal	0.0000	1.147
21	2-Propanol	0.0000	1.185	21	Furfural	0.0001	1.080
22	Ethanol	0.0000	1.219	22	Hexanal	0.0037	1.087
23	2-Ethylhexanol	0.0000	1.234	23	(*E*)-2-Pentenal	0.0000	1.132
24	2-Methyl-1-butanol	0.0002	1.175	24	(*E*)-2-Hexenal	0.0000	1.153
25	Methyl butanoate	0.0000	1.178	25	3-Methyl-butanal	0.0034	1.000
26	Ethyl Acetate	0.0000	1.173	26	Benzaldehyde	0.0000	1.125
27	Butyl acetate	0.0003	1.174	27	Linalool	0.0000	1.058
28	Ethyl 2-methylbutanoate	0.0064	1.127	28	2-Furanmethanol	0.0044	1.046
29	Dimethyl sulfide	0.0000	1.173	29	1-Butanol	0.0043	1.100
30	Isobutyric acid	0.0076	1.008	30	Methanol	0.0030	1.121
31				31	1-Pentanol	0.0000	1.101
32				32	Ethanol	0.0245	1.012
33				33	Methionol	0.0000	1.049
34				34	2-Methyl-1-butanol	0.0001	1.123
35				35	Geranyl acetate	0.0021	1.029
36				36	Phenylethyl acetate	0.0016	1.018
37				37	Butyl acetate	0.0029	1.058
38				38	2-Acetylfuran	0.0003	1.153
39				39	Dimethyl sulfide	0.0046	1.014
40				40	2-Methylbutanoic acid	0.0000	1.059

**Table 3 molecules-28-06865-t003:** Odor activity values (OAV) of *Schisandra chinensis* branch exudates and dried fruit of different fruit colors.

Compound	W1	W2	W3	W4	W5	W6
β-Ocimene	1.03 ± 0.45	1.49 ± 0.19	1.37 ± 0.20	45.79 ± 0.99	55.01 ± 1.03	11.73 ± 1.65
Limonene	0.05 ± 0.02	0.07 ± 0.01	0.06 ± 0.01	2.12 ± 0.06	2.80 ± 0.01	0.89 ± 0.04
Myrcene	0.30 ± 0.12	0.38 ± 0.02	0.37 ± 0.02	8.69 ± 0.22	13.03 ± 0.27	5.59 ± 0.38
β-Pinene	0.60 ± 0.13	0.82 ± 0.07	0.71 ± 0.07	16.59 ± 0.30	16.54 ± 0.09	9.00 ± 1.04
α-Pinene	0.22 ± 0.05	0.23 ± 0.03	0.20 ± 0.03	8.40 ± 0.36	10.16 ± 0.02	4.40 ± 0.82
6-Methyl-5-hepten-2-one	1.83 ± 0.24	1.98 ± 0.13	1.82 ± 0.08	38.78 ± 1.78	31.09 ± 0.22	40.29 ± 1.98
Acetone	0.70 ± 0.01	1.00 ± 0.06	0.31 ± 0.00	1.80 ± 0.12	1.86 ± 0.07	1.64 ± 0.02
3-Octanone	57.57 ± 2.57	59.15 ± 4.45	33.79 ± 0.85	16.73 ± 2.97	37.75 ± 0.82	6.01 ± 0.81
Nonanal	30.62 ± 4.34	33.94 ± 4.35	78.67 ± 1.49	69.44 ± 3.66	69.73 ± 2.50	74.09 ± 3.53
(*E*)-2-Hexenal	56.61 ± 30.14	30.24 ± 2.03	30.49 ± 3.33	929.23 ± 98.24	423.44 ± 40.04	221.68 ± 15.89
2-Methylbutanal	1.32 ± 0.02	1.11 ± 0.01	1.93 ± 0.01	2.65 ± 0.42	2.96 ± 0.17	2.12 ± 0.05
3-Methyl-butanal	3132.27 ± 20.20	2364.55 ± 111.13	523.32 ± 8.24	609.15 ± 84.54	453.54 ± 12.10	785.83 ± 85.75
Linalool	39.76 ± 5.15	41.38 ± 0.77	36.48 ± 2.82	291.66 ± 19.67	351.46 ± 16.55	76.44 ± 7.13
1-Butanol	0.66 ± 0.01	0.52 ± 0.04	0.61 ± 0.04	1.42 ± 0.18	0.94 ± 0.03	1.06 ± 0.06
1-Propanol	1.57 ± 0.01	1.38 ± 0.10	0.64 ± 0.00	1.71 ± 0.15	1.93 ± 0.19	1.56 ± 0.10
Ethanol	1.19 ± 0.02	1.96 ± 0.09	1.17 ± 0.01	0.58 ± 0.05	0.96 ± 0.10	0.93 ± 0.20
Phenylethyl acetate	12.30 ± 2.99	14.39 ± 0.84	14.59 ± 1.40	27.10 ± 3.52	46.46 ± 8.64	16.88 ± 1.57
Methyl butanoate	1.63 ± 0.05	1.36 ± 0.07	1.05 ± 0.03	3.37 ± 0.52	2.55 ± 0.20	2.73 ± 0.09
Butyl acetate	2.32 ± 0.17	2.76 ± 0.05	3.10 ± 0.06	1.18 ± 0.13	1.62 ± 0.29	2.25 ± 0.21
1,8-Cineole	11.22 ± 2.91	14.33 ± 1.45	13.03 ± 1.37	326.88 ± 1.64	361.18 ± 24.93	289.20 ± 12.48
Dimethyl sulfide	107.89 ± 0.51	102.60 ± 4.19	67.71 ± 1.84	241.33 ± 3.31	196.30 ± 77.59	48.91 ± 3.98
2-Methylbutanoic acid	0.00 ± 0.00	0.01 ± 0.00	0.00 ± 0.00	2.34 ± 0.04	3.55 ± 0.20	1.47 ± 0.24
Isobutyric acid	0.00 ± 0.00	0.00 ± 0.00	0.00 ± 0.00	1.61 ± 0.34	1.23 ± 0.19	2.10 ± 0.33
Pentanoic acid	0.00 ± 0.00	0.00 ± 0.00	0.00 ± 0.00	28.00 ± 1.81	53.09 ± 13.00	68.39 ± 14.38

## Data Availability

All related data and methods are presented in this paper. Additional inquiries should be addressed to the corresponding author.

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
