# Peer review of "Analysis of Volatile Components in Dried Fruits and Branch Exudates of Schisandra chinensis with Different Fruit Colors Using GC-IMS Technology"

_molecules, 2023, doi:10.3390/molecules28196865_

Round 1

Reviewer 1 Report

This study was conducted to analyze and evaluate the volatile components of Schisandra chinensis as a traditional medicinal plant. There are similar studies on the volatile components of Schisandra chinensis. Together with these prior studies, this article seems to be a good study identifying the components of 3 different colors of Schisandra fruit. However, there are many sentences that require correction and additional explanation.

 Over all

-      It is recommended to write the genus and species names of plants in italics.

-      Some parts were written in command form rather than passive voice, and some parts were written in present or future tense rather than past tense. They must be corrected.

-      The numbering in the section 2 (materials an methods) and section 3 (results and discussion) must be corrected.

-      Figures: Abbreviations should be defined the first time they appear in the first figure.

Line 36-38:  Please indicate corresponding references.

Line 83:  What year were the fruits harvested?

Line 84-85:  Drying methods are very important for studying volatile compounds. In this article, only the drying method is described as “dried in shade”. The degree of drying, or at least the drying temperature and period must be indicated.

The quality of English language is not bad. However, as a scientific paper, there are many sentences that require correction.

Reviewer 2 Report

This is an extensive and interesting research, but in my opinion some important corrections should be made.

The Introduction needs to be improved in terms of data related to Schisandra chinensis, as well as advantages/disadvantages of the GC-IMS technique. The aim of the work is not clearly stated.

In Materials and methods it is unclear how the fruit samples were dried. It is necessary to explain in more detail the pre-treatment of branches and dried fruits. There is no data on the operating conditions of the detector and how the spectra were acquired. It is necessary to describe in detail all experimental procedures and experimental conditions.

Important part of this manuscript is application of different statistical approaches (PCA, OPLS-DA, clustering) and there are lack of information how this analysis were performed. The resolution of images 7 to 9 needs to be improved and the discussion related to image 7 is completely missing.

Considering the results presented in this manuscript, the similarities and differences between the analyzed samples need to be described in more detail.

In my opinion, considerations given from line 353 to line 366 should be move in discussion.

Minor corrections of the English language and grammar are needed.

Round 2

Reviewer 2 Report

The authors addressed to all of comments, but in my opinion not in the satisfactory way.

In Materials and methods section, subsection 3.2.2. (Lines 326-330) the added part is completely inappropriate. The authors either did not understand the comments or did not pay enough attention to what was written. It is generally known how to program the operation of an instrument and how to start the instrument itself. My question was about the operating conditions of the detector, such as electric field strength, total voltage, etc. If the comment is not clear, see: https://doi.org/10.3390/foods8100460  or

https://doi.org/10.3390/molecules26185457http.

  Same comment is for next sentence: „After the instrument finished the test, the data were automatically transferred to the computer via a data cable.‟ (Lines 341-342). Delete this sentence, because software is a data collection and processing tool.

Moderate corrections of the English language and grammar are needed.
